# Early Childhood Stimulating Environment Predicts Later Childhood Resilience in an Indian Longitudinal Birth Cohort Study

**DOI:** 10.3390/children9111721

**Published:** 2022-11-09

**Authors:** Beena Koshy, Manikandan Srinivasan, Sowmiya Gopalakrishnan, Venkata Raghava Mohan, Rebecca Scharf, Sushil John, Rachel Beulah, Jayaprakash Muliyil, Gagandeep Kang

**Affiliations:** 1Developmental Paediatrics, Christian Medical College, Vellore 632004, India; 2Wellcome Research, Christian Medical College, Vellore 632004, India; 3Community Health, Christian Medical College, Vellore 632004, India; 4Centre for Global Health, University of Virginia Children’s Hospital, Virginia, VA 22903, USA; 5Low Cost Effective Care Unit, Christian Medical College, Vellore 632004, India

**Keywords:** early childhood, cognition, home environment, resilience

## Abstract

Background: Human resilience is dynamic and generated through myriad interactions starting from early childhood. Resilience can improve quality of life and longevity. Objective: The current analysis evaluates childhood resilience at 9 years of age and its associations with concurrent cognition and early childhood factors, using longitudinal follow-up of a community birth cohort in Vellore, south India. Methods: Resilience was assessed using the Child and Youth Resilience Measure and cognition utilizing the Malin’s Intelligence Scale for Indian Children. Early childhood variables included growth stunting, maternal depression scores, home environment scores, and socio-economic status (SES) at 2 years of age. Statistical evaluation included bivariate analysis with further multi-variate regression for each resilience domain and the total resilience score. Findings: Out of 251 newborns recruited into the original birth cohort, 205 (81.7%) children were available for the 9-year follow-up. Mean (SD) scores in individual, caregiving, and context domains of resilience were 26.34 (3.2), 19.52 (1.6), and 20 (1.8) respectively. Individual resilience domain was associated with verbal cognition scores at 9 years of age (0.07, *p* = 0.019) and total home environment scores (0.16, *p* = 0.027) at 2 years of age, when adjusted for stunting, maternal depression, and SES. The total resilience score was associated only with concurrent verbal intelligence (0.08, *p* = 0.026) after adjustment with early childhood factors. Analysis of individual home environment factors showed that daily stimulation was associated with the individual domain of resilience. Conclusions: Follow-up of an Indian birth cohort showed that in addition to concurrent cognitive abilities, childhood resilience was related to early childhood stimulating home environment. Promoting optimal stimulating home environments in low-resource settings to nurture holistic childhood development including mental health is essential.

## 1. Background

Human resilience is dynamic and generated through myriad interactions starting from early childhood. Resilience is defined as the ability to overcome adversity and can help to improve one’s quality of life and longevity [1,2]. Psychologically the construct of resilience can be ascribed to continuing competency under threat, good or above-expected outcomes despite high-risk exposure, and recovery from distress and trauma [3]. Resilience is important not just from an individual perspective, but also for all humankind as human resilience helps to succeed over risks, threats, and disasters [3]. Salient components of the course of resilience development include adversity exposure, overcoming obstacles, and successful outcomes [4].

Childhood resilience is of interest as early childhood factors can help in shaping resilience for life. Childhood is a critical phase for resilience from its contentious nature-nurture origins, its associations with the child’s evolving cognitive, emotional, and psycho-social development, and its modulations from micro and macro-environments of early life [1,3]. In the context of early childhood poverty and its inter-generational transmission, resilience can aid in overcoming this early childhood adversity for better outcomes and possible upward social mobility [3]. Moreover, effective childhood resilience adaptation with supportive family and community networks can result in better mental health and well-being for life [5,6].

Globally, there is a renewed interest in resilience and mental health, especially during current times of pandemics, conflicts, and wars. The United Nations Sustainable Development Goal (SDG) 3 aims to support mental health for all in order to achieve global health [7]. An effective childhood resilience development can aid in an individual’s well-being and mental health as well as coping, adaptation, and survival mechanisms, which in turn aid to develop resilient community and global populations, which can deal with risks and harness resources better [8,9,10].

For resilience development, the interplay between vulnerability and protective factors, and intrinsic and extrinsic influences determines successful outcomes after risk or stress exposure [11]. Protective factors augmenting resilience development can include positive community and family cultural and relational facets along with an individual’s intrinsic characteristics such as intelligence, temperament, and emotional well-being [1,4,11]. Early childhood home environment including its physical and relational components are known to be associated with childhood resilience maturation [12], but resilience studies have not kept abreast with evolving home environments over time, especially in different cultural and income settings. Protective home environmental factors could be enriching and stimulating home environments [13,14], inter-connected familial systems [5], stable marriages [9], female non-single caregivers [15], responsive and interactive parenting [12] and caregiver support [16]. Despite a good body of work on childhood resilience, the complexity of its origins, associations, contributing factors, and evolutions have not been fully understood. 

Socio-cultural factors along with community and family support influence resilience development as discussed, thus necessitating exploration of the construct of resilience in different cultural milieux [16]. Literature exploring resilience is sparse in India, with most published articles exploring resilience in adults [17,18,19] or in children with neurodevelopmental disorders [20]. Network analysis of resilience across 14 countries including India highlighted the influence of supportive caregivers on adolescent resilience [16]. In this background, the current study evaluates childhood resilience at 9 years of age and its associations with concurrent cognition and early childhood factors, using longitudinal follow-up of a community birth cohort in Vellore, south India. It is hypothesized that early childhood home environments will be predictively associated with later childhood resilience.

## 2. Methodology

### 2.1. Settings and Subjects

This analysis utilized the follow-up of a birth cohort registered for the ‘Etiology, Risk Factors and Interactions of Enteric Infections and Malnutrition and the Consequences for Child Health and Development’ (MAL-ED) Network, a prospective multinational longitudinal cohort study in 8 countries across the world [21]. The site in India was based in Vellore, South India compiling of 8 adjacent heavily populated urban slum settlements of low-and-middle income backgrounds [22]. Previous publications from the same cohort site have highlighted study specifics such as population characteristics, recruitment process, inclusion and exclusion criteria and follow-up [23,24,25,26,27]. 

The original MAL-ED cohort enrolment between February 2010 and February 2012 enlisted 251 newborns in the Indian site. Children were subsequently followed up as per the protocol. The present analysis evaluates associations between early childhood factors at 2 years of age and childhood resilience at 9 years of age. The age of 2 years was selected as it marks the conclusion of the first 1000 days of individual life including the in-utero period [28]. The Institutional Review Board (IRB) of Christian Medical College (CMC) Vellore reviewed and approved the initial cohort recruitment stage and later follow-ups. Parental informed consent was obtained in written format before each recruitment and for the 9-year enrolment, appropriate child assent was also obtained. 

### 2.2. Measures

We measured length at 2 years of age with an infantometer to the nearest possible cm. Stunting was defined as <−2 standard deviation using the Multicentre Growth Reference Study (MGRS) standards [29]. 

#### 2.2.1. The Child Youth Resilience Measure (CYRM)

Resilience was analyzed using the Child and Youth Resilience Measure (CYRM), modified for use in the mid-childhood (5–9 years) [30]. The 3 sub-scales included Context, Caregiver, and Individual resilience. The CYRM had 26 items with a 3-point Likert response scale (Yes, Sometimes, No). The subscale of individual resilience evaluated personal skills including social interaction and coping, peer support and behavioral modulatory skills. The subscale of caregiver component analyzed perceptions of physical and psychological caregiving. The context subscale had components of spiritual, education and cultural participation. For uniformity, questions were read to children by a single trained psychologist with their responses recorded at 9 years of age.

#### 2.2.2. The Malin’s Intelligence Scale for Indian Children (MISIC)

The Malin’s Intelligence Scale for Indian Children (MISIC) was modelled on the original Wechsler Intelligence Scale for Children (WISC) and had verbal and performance subscales [31]. Age range for this measure was 6–16 years. A single community psychologist administered MISIC to children in the community clinic setting at 9 years of age. We used chronological and mental child ages to calculate cognitive standardized scores and verbal, performance, and total intelligence quotients. Verbal intelligence quotient (VIQ) was drawn using verbal subscales and performance intelligence quotient (PIQ) utilizing components of performance subscales. VIQ measured verbal ability of the child including information, comprehension, narration, vocabulary etc. PIQ measured non-verbal abilities such as picture completion, block design, object assembly etc. 

#### 2.2.3. The WAMI for Socio-Economic Status

A composite tool including factors such as (1) access to improved Water and sanitation, (2) Assets, (3) Maternal education and (4) total household Income (WAMI) was utilized to analyze socio-economic status (SES) at 2 years of age in the MAL-ED cohort [32]. The final score from all the 4 variables were transformed into a standardized score (0–1).

#### 2.2.4. The Home Observation for the Measurement of the Environment (HOME) Scale

The Home Observation for the Measurement of the Environment (HOME) analyzed the overall support, the child received at home. The modified version used in the current analysis had six subscales: Avoidance of restriction/punishment, Appropriate play materials, Organization of the environment, Responsiveness to one’s parent, Parental involvement, and Variety in daily stimulation [23]. The HOME was administered at 2 years of child’s age by a trained community social worker after observing the environment at home for close to 1 h with additional information from interviewing caregiver/s [21,33]. 

#### 2.2.5. The Self Reporting Questionnaire-20

Maternal psychological disturbances were assessed using the Self Reporting Questionnaire-20 (SRQ-20) of the World Health Organization (WHO). The SRQ-20 tool was envisaged to be used in low-resource settings [34] A trained community psychologist administered this measure. For the current analysis, we used the assessment conducted at 2 years of child’s age.

### 2.3. Data Entry and Analysis

The MAL-ED cohort had a Data Co-ordinating Centre managing data entry, checks and corrections. In Vellore, field supervisor validated all paper forms before data entry and information was enrolled into the electronic database using a double entry database system [21]

### 2.4. Statistical Analysis

From the original cohort, baseline characteristics and anthropometry of children were delineated. Maternal depression and HOME inventory scores at 2 years of age, and cognition scores (VIQ and PIQ) at 9 years of age were presented as continuous variables. Children with WAMI scores below 33rd percentile were classified into low SES and those with WAMI scores ≥ 33rd percentile were classified as high SES. Resilience scores were summarized as mean (standard deviation) and median (interquartile range) under the domains of individual, caregiving, context, and total resilience scores. To assess the association between various domains of resilience scores and cognition scores at 9 years along with early childhood factors, bivariate and further multivariate regression analysis were performed. In regression analysis, domain-wise resilience scores were considered as the outcome variable and cognition scores at 9 years, HOME inventory scores, maternal depression scores, stunting and SES status at 2 years as predictors. Significant predictors for domain-wise resilience scores in bivariate analysis were considered for multivariate analysis. Multivariate regression model was checked for heteroscedasticity and R^2^ value was used to assess model fitness. Beta co-efficients with 95% confidence interval were described and *p*-value < 0.05 was deemed as the level of significance. We used Stata version 13 (StataCorp. 2013. Stata Statistical Software. Release 13. College Station, TX, USA: StataCorp LP) software for all statistical analysis.

## 3. Results

251 newborns were recruited in this birth cohort between 2010-12, of which 205 (81.67%) were available for follow-up at 9 years, with no significant difference from the original birth cohort [25]. At 2 years of age, 44.50% (101/228) of children were stunted. Mean (SD) score of HOME inventory and maternal depression scale at 2 years was 40.23 (3.43) and 3.88 (3.21) respectively. A total of 71 out of 228 children (31.14%) belonged to low SES at 2 years of follow-up (Table 1). Mean VIQ and PIQ (SD) scores at 9 years were 94.03 (9.71) and 91.49 (13.01) respectively. 

At 9 years, mean (SD) scores of the individual, caregiving, context, and total resilience were 26.34 (3.19), 19.52 (1.64), 20 (1.80) and 65.86 (4.69) respectively (Appendix A). Internal consistency of items of the questionnaire was tested using Cronbach’s alpha and alpha co-efficient was found to be 0.54. In the bivariate analysis, both individual and total resilience domains showed positive association with increase in VIQ and HOME inventory scores, and a negative association with increase in maternal depression scores. Higher PIQ scores at 9 years and low SES background at 2 years were the other significant predictors of individual domain of resilience scores. Analyzing caregiving domain of resilience scores at 9 years, a negative association was observed with maternal depression scores in the early childhood. However, none of the predictors were found to be associated with the context domain of resilience scores (Table 2).

Higher VIQ scores [0.07 (95% CI: 0.01–0.13)] and HOME inventory scores [0.16 (95% CI: 0.02–0.30)] showed positive association with individual domain of resilience scores. Also, the model considering total resilience scores as outcome showed a positive association with higher VIQ scores [0.08 (95% CI: 0.01–0.15)] (Table 3). Further analysis with the addition of specific domains of HOME inventory scores into the model showed that daily stimulation available at home [0.32 (95% CI: 0.04–0.60)] and environment related factors in home [0.47 (95% CI: 0.02–0.93)] during the early childhood were positively associated with individual domain and total resilience scores respectively at 9 years (Table 4).

## 4. Discussion

The current analysis evaluated childhood resilience and its association with concurrent cognition and early childhood influences utilizing a birth cohort follow-up in a low-and-middle income urban setting in south India and showed that individual component of resilience was associated with early childhood home environment. At least one of the 3 domains of resilience namely individual, caregiving, and context domains was associated with concurrent verbal and performance cognition at 9 years, and mother’s depression, home environment, and SES at 2 years of age. While the individual component of resilience evaluated personal coping and social interactions, the caregiver domain explored the child’s perceptions of parenting, and the context domain her engagement in education and cultural environment. In the adjusted analysis, the child’s individual resilience domain was associated with 9-year VIQ scores and 2-year home environment scores. Total resilience score was associated with only concurrent VIQ, while caregiving and context domains did not have any association, when adjusted for early childhood factors. Evaluating components of early home environment that could have contributed to resilience, daily stimulation was found associated with the individual domain of resilience and home environment with total resilience. 

More than 80% of children recruited in the original birth cohort were available for follow-up 9 years later in the current analysis and the 9-year group did not significantly differ from the original birth cohort as indicated in a previous publication [25]. Few studies have explored resilience in birth cohort follow-up settings and the present study adds evidence to resilience literature in terms of its early childhood home environment association in low-and-middle income south Indian milieu. A long-term follow-up Hawaiin study found that vulnerable children developed high-risk behavior in adolescence but eventually developed positive outcomes by mid-life predominantly modulated by continuing educational and vocational achievements [9]. The Philadelphia Neurodevelopmental Cohort (PNC) follow-up had shown that traumatic stressful events were associated with higher lifetime psychopathology including anxiety and psychosis [35,36,37].

The predictive relationship of early childhood stimulating home environment with later childhood resilience as shown in this study is in concurrence with the available literature. Enriching, encouraging, and stimulating home environments form the foundation of a positive behavioral profile including resilience [13,14,38]. Early supportive family home environments including family relationships are protective factors to develop adaptation and resilience [11]. Home environments where the child spends the maximum time can be broadly divided into physical (safety, overcrowding, physical facilities) and relational (sensitive and responsive caregiving) environments [39]. Previous analyses from the same Indian birth cohort have shown a close relationship between early home environment and SES [23] and a positive association between nurturing home environment and development trends in early childhood between 6 and 36 months of age [24]. The current study adds extra impetus to work on early childhood home environments by demonstrating its association with later childhood resilience even after corrections with SES, maternal factors, and concurrent cognition. Optimal early childhood responsive and organized care can help to recover from an early insult such as prematurity to develop normal cognition and early resilience [12].

Maternal factors such as well-being, education, and cognitive abilities were associated with the relational home environment in early childhood in our previous analysis [23] in concurrence with literature findings of the significance of maternal caregiving and nurture in early childhood [12,40]. Responsive and sensitive caregiving can be a moderator and strong protective factor to developing resiliency in the face of adversity such as prematurity and poverty [12,40]. In our current study, maternal depression at 2 years of age was negatively associated with individual, caregiving, and total scores of childhood resilience in bivariate analysis stressing the need to improve maternal well-being, well recognized and highlighted by the WHO [41].

Resilience development entails exposure to risks within the supportive framework of families and communities. Protective and vulnerability factors modulate this response within the attributes of the child, family, and community [4,11]. Protective factors include early family relationships [11], safer, less crowded homes [12], well-developed support systems [5] continuing educational and vocational opportunities, marriage to a stable partner and religious involvement [9]. Social ecologies including interconnected support networks of families and communities, and cultural frameworks play an important bedrock for successful resilience development [1,5,16,19]. The urban slum setting of the current birth cohort has many challenges including infections, poor sanitation, and sub-optimal resources, but has an inter-connected familial and social system supportive of each other. As there is a call to understand resilience across cultures [42], the current analysis highlights the association of early childhood home environment especially daily stimulation with childhood resilience in a low-and-middle income setting in south India.

Successful adaptation including resilience is integral to holistic human development in an everchanging environment of challenges [2,10]. Resilience is multi-dimensional, integrative, and portable across system levels [2,5], enhancing positive thinking, functioning, flexibility, and hardiness which in turn facilitate better mental health, optimal resource harness, and better quality of life [6,10]. The resilience of individuals and communities can help to resolve and overcome adversity including poverty, tensions, and conflicts [3,42]. 

The Indian government has taken many commendable initiatives to improve childhood nutrition, growth, and development including ‘Mission Poshan 2’ also known as the National Nutrition Mission to improve the nutrition of pregnant women, mothers, and children, ‘Mission Shakti’ to empower women and children and ‘Mission Vatsalya’ to ensure healthy childhood [43]. In addition to the integration of these schemes at community levels, optimizing home environments including responsive caregiving with the utilization of local and home-based resources as envisaged by the WHO is needed [41].

Limitations to the current analysis include a comparatively smaller sample size with minimal loss to follow-up. Strengths of the MAL-ED India cohort include dense data granularity in the baseline early childhood, standardized assessments including those of growth, home environment, and SES with acceptable quality assurance, and India-specific cognitive assessment in childhood.

## 5. Conclusions

This follow-up study of an Indian birth cohort showed that in addition to concurrent cognitive abilities, childhood resilience was related to early childhood stimulating home environment. Promoting optimal stimulating home environments in low-resource settings to nurture holistic childhood development including mental health is essential.

## Figures and Tables

**Table 1 children-09-01721-t001:** Sociodemographic characteristics and HOME inventory scores of MAL-ED cohort at enrolment, 2 and 9 years of follow-up.

	Enrolment (*n* = 251)	2 Years (*n* = 228)	9 Years (*n* = 205)
**Gender**			
Male	113 (45)	105 (46.05)	96 (46.83)
Female	138 (55)	123 (53.95)	109 (53.17)
**Socioeconomic status**			
Low (WAMI < 33rd percentile)	71 (30.2)	71 (31.14)	-
High (WAMI ≥ 33rd percentile)	164 (69.79)	157 (68.86)	
**Stunting status ^#^**			
Stunted (HAZ < −2 SD)	41 (16.33)	101 (44.5)	22 (10.73)
Not Stunted (HAZ ≥ −2 SD)	210 (83.67)	126 (55.5)	183 (89.27)
**Maternal depression scores**, median (IQR)	-	3 (1–6)	-
**HOME Inventory Scale, mean (SD)**			
Total domain scores	-	40.23 (3.43)	-
Emotional domain		10.98 (0.16)	
Domain on provision of play materials		2.38 (0.49)	
Daily stimulation domain		5.86 (1.59)	
Home-related environmental domain		9.58 (1.54)	

^#^ Information on stunting status was available for 227/228 children at 2-years follow-up. MAL-ED—The ‘Etiology, Risk Factors and Interactions of Enteric Infections and Malnutrition and the Consequences for Child Health and Development’ Study. IQR—Interquartile range.

**Table 2 children-09-01721-t002:** Bivariate analysis of various domains of resilience scores at 9 years with concurrent cognition and early childhood factors in the MAL-ED cohort (*n* = 205).

Predictors	Unadjusted Beta Co-Efficients with 95% Confidence Intervals			
	Individual Domain	Caregiving Domain	Contextual Domain	Total Resilience Scores
Verbal IQ scores at 9 years	**0.08 (0.04–0.13)**	0 (−0.02–0.02)	0.02 (−0.01–0.04)	**0.09 (0.03–0.16)**
Performance IQ scores at 9 years	**0.04 (0.01–0.08)**	0 (−0.02–0.01)	0 (−0.02–0.02)	0.04 (−0.01–0.09)
Stunted at 2 years *	0.18 (−0.71–1.07)	0.26 (−0.18- 0.72)	−0.06 (−0.56–0.44)	0.39 (−0.92–1.69)
Maternal depression scores at 2 years	**−0.15 (−0.29–−0.02)**	**−0.08 (−0.15–−0.14)**	0 (−0.07–0.08)	**−0.23 (−0.43–−0.04)**
HOME inventory scores at 2 years	**0.23 (0.10–0.35)**	0.06 (−0.01–0.12)	−0.02 (−0.09–0.06)	**0.27 (0.08–0.46)**
Low socioeconomic status at 2 years * (WAMI < 33rd percentile)	**−1.02 (−1.96–−0.08)**	0 (−0.49–0.48)	−0.25 (−0.79–0.29)	−1.27 (−2.66–0.12)

Beta co-efficients with significance level of *p* < 0.05 were presented in bold fonts. MAL-ED—The ‘Etiology, Risk Factors and Interactions of Enteric Infections and Malnutrition and the Consequences for Child Health and Development’ Study. IQ—Intelligence quotient. HOME—The Home Observation for the Measurement of the Environment. WAMI—Water and sanitation, Assets, Maternal education and total household Income. * Data points on stunting status and socioeconomic status are available only for 204/205 children. In the multivariate analysis, only individual domain and total resilience scores were considered as outcomes.

**Table 3 children-09-01721-t003:** Multivariate analysis of individual and total domains of resilience scores at 9 years with concurrent cognition and early childhood factors in MAL-ED cohort (*n* = 204 *).

Predictors	Adjusted Beta Co-Efficients with 95% Confidence Intervals	
	Individual Domain of Resilience Scores	Total Resilience Scores
Verbal IQ scores at 9 years	**0.07 (0.01–0.13)**	**0.08 (0.01–0.15)**
Performance IQ scores at 9 years	0 (−0.05–0.04)	-
Maternal depression scores at 2 years	−0.10 (−0.24–0.04)	−0.18 (−0.38–0.03)
HOME inventory scores at 2 years	**0.16 (0.02–0.30)**	0.17 (−0.04–0.37)
Low socioeconomic status at 2 years (WAMI < 33rd percentile)	0.03 (−1.0–1.05)	0 (−1.54–1.54)

R^2^ value for individual domain of resilience scores and total resilience scores was 8.25% and 5.75% respectively. Beta co-efficients with significance level of *p* < 0.05 were presented in bold fonts. MAL-ED—The ‘Etiology, Risk Factors and Interactions of Enteric Infections and Malnutrition and the Consequences for Child Health and Development’ Study. IQ—Intelligence quotient. HOME—The Home Observation for the Measurement of the Environment. WAMI—Water and sanitation, Assets, Maternal education and total household Income. * Data points on socioeconomic status is available only for 204/205 children and thus 204 children were included in the model.

**Table 4 children-09-01721-t004:** Multivariate analysis of specific domains of resilience scores with domains of HOME in children of MAL-ED cohort (*n* = 204 *).

Predictors	Adjusted Beta Co-Efficients with 95% Confidence Intervals	
	Individual Domain of Resilience Scores	Total Resilience Scores
Verbal IQ scores at 9 years	**0.07 (0.01–0.13)**	**0.08 (0.01–0.14)**
Performance IQ scores at 9 years	0 (−0.04–0.04)	-
Maternal depression scores at 2 years	−0.12 (−0.25–0.02)	−0.18 (−0.38–0.03)
Daily stimulation domain of HOME inventory scores at 2 years	**0.32 (0.04–0.60)**	-
Environment related domain of HOME inventory scores at 2 years	-	**0.47 (0.02–0.93)**
Low socioeconomic status at 2 years (WAMI < 33rd percentile) *	0.14 (−0.86–1.14)	−0.09 (−1.62–1.43)

R^2^ value for individual domain of resilience scores and total resilience scores is 8.29% and 6.55% respectively. Beta co-efficients with significance level of *p* < 0.05 were presented in bold fonts. HOME—The Home Observation for the Measurement of the Environment. MAL-ED—The ‘Etiology, Risk Factors and Interactions of Enteric Infections and Malnutrition and the Consequences for Child Health and Development’ Study. IQ—Intelligence quotient. WAMI—Water and sanitation, Assets, Maternal education, and total household Income. * Data points on socioeconomic status are available only for 204/205 children and thus 204 children were included in the model.

## Data Availability

MAL-ED data from all sites are deposited in the https://clincpidb.org website and can be accessed after appropriate permissions. The 9-year dataset is available with the corresponding author.

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
