# Peer review of "Early Childhood Stimulating Environment Predicts Later Childhood Resilience in an Indian Longitudinal Birth Cohort Study"

_children, 2022, doi:10.3390/children9111721_

Round 1
Reviewer 1 Report
Paper title: “Early childhood stimulating environment predicts later child-2 hood resilience in an Indian longitudinal birth cohort study”
Thank you for your paper, I do believe it will make a valuable contribution to the discussion concerning resilience using an longitudinal birth cohor study. Your paper is well written, and interesting. The article introduces the resilience and the factors. However I do suggest some changes and points to address below:
First, the introduction is too short. Because the reader knows so little about the resilience, the author needs to introduce more about it. For example, the importance of resilience; the importance of resilience in childhood stage; the impact of resilience on the children’s future; and the literature about the factors that influence resilience.
Secondly, there is some repetition in THEORETICAL BACKGROUND section and the last paragraph of the introduction section.
Secondly, I think the author should add some tables, such as table 1 of the characteristics of child and family, from this table we can see the gender, average age, health, maternal education and age, the asset of the family and so on. , and table 2 of the description of the home environment. We can know the read, sing, toys and other information about the home environment. Table 3 of the childhood development.
Third , the author should add the heterogeneity and the mechanism analysis. I think these analyses is interesting and is helpful to the literatures.
Author Response
To
The reviewers
Children
Re: Manuscript ID: children-1973057
Title: Early childhood stimulating environment predicts later childhood resilience in an Indian longitudinal birth cohort study
Dear reviewer,
Thank you for reviewing our manuscript titled “Early childhood stimulating environment predicts later childhood resilience in an Indian longitudinal birth cohort study”. Authors’ responses are provided below
|
Comments |
Authors’ response |
Page number |
|
Reviewer 1 |
|
|
|
1. First, the introduction is too short. Because the reader knows so little about the resilience, the author needs to introduce more about it. For example, the importance of resilience; the importance of resilience in childhood stage; the impact of resilience on the children’s future; and the literature about the factors that influence resilience. |
Thank you. Introduction is rewritten as different paragraphs detailing 1. Resilience 2. Childhood resilience 3. Resilience evolution for child and global health and future 4. Factors influencing resilience 5. Indian literature |
Pg 6-7 |
|
2. Secondly, there is some repetition in THEORETICAL BACKGROUND section and the last paragraph of the introduction section |
Thank you Introduction edited |
Pg 6-7 |
|
3. Secondly, I think the author should add some tables, such as table 1 of the characteristics of child and family, from this table we can see the gender, average age, health, maternal education and age, the asset of the family and so on. , and table 2 of the description of the home environment. We can know the read, sing, toys and other information about the home environment. Table 3 of the childhood development. |
Thank you. As suggested, we have added table 1 with sociodemographic characteristics and HOME inventory scores at enrolment, 2 and 9 years of follow-up. IQ scores recorded at 9 years has already been presented in the Results as text. |
Table 1 |
|
4. Third, the author should add the heterogeneity and the mechanism analysis. I think these analyses is interesting and is helpful to the literatures |
Thank you. We have checked for heteroscedasticity for the regression model using rvf plot and this information has been added in statistical analysis section. |
Pg 11 |
Thanking you
Yours sincerely
Authors
Reviewer 2 Report
This article on resilience in a particular Indian sample studied longitudinally is part of a series on the cohort. The results emphasize the early home environment as a contributing factor to the outcome. The methodology is comprehensive, but I have some concerns with its report, the alpha level used, and the writing. About the methods, the measures used do not include reliability data, such as Chronbach’s alpha. About statistics, p = .05 for so many measures and stats does not accommodate Type 1 error. But using less than p = .5 will eliminate much of the significant findings, so this is an editorial call. About writing, I understood every sentence, but many lacked basics – omitting the word “the”, using singular instead of plural, or vice versa. Commas are missing. One correction -- milieux is the plural. Also, multivariable should be multivariate. An English speaker should carefully proof, please. About definitions, use PIQ and VIQ, SES, and others in the text after first mention, where they should be defined. No need for an abbreviation Table. What is IQR? About the hypotheses, what were they, if any, and what aspects of the literature justified them? About the discussion, at the beginning, more could be written on the specific results and their interpretations, instead of writing generally about resilience. This would mirror the specific hypotheses made.
Author Response
To
The reviewers
Children
Re: Manuscript ID: children-1973057
Title: Early childhood stimulating environment predicts later childhood resilience in an Indian longitudinal birth cohort study
Dear reviewer,
Thank you for reviewing our manuscript titled “Early childhood stimulating environment predicts later childhood resilience in an Indian longitudinal birth cohort study”. Authors’ responses are provided below
|
Reviewer 2 |
|
|
|
1. About the methods, the measures used do not include reliability data, such as Chronbach’s alpha. |
Thank you. Cronbach’s alpha of the items of the questionnaire was found to be 0.54 and the same has been added to the Results section. |
Pg 14 |
|
2. About statistics, p = .05 for so many measures and stats does not accommodate Type 1 error. But using less than p = .5 will eliminate much of the significant findings, so this is an editorial call. |
Thank you. In the regression analysis, p-value < 0.05 was considered as the level of statistical significance, in order to rule out associations that are merely due to chance. We are happy to provide any further clarifications on this. |
|
|
3. About writing, I understood every sentence, but many lacked basics – omitting the word “the”, using singular instead of plural, or vice versa. Commas are missing. One correction -- milieux is the plural. Also, multivariable should be multivariate. An English speaker should carefully proof, please. |
Thank you. We have re-edited the whole manuscript. |
|
|
4. About definitions, use PIQ and VIQ, SES, and others in the text after first mention, where they should be defined. |
Thank you. Corresponding changes have been made |
Pg 10 |
|
5. No need for an abbreviation Table. |
Thank you. Table removed |
|
|
6. What is IQR ? |
IQR refers to inter-quartile range. This abbreviation has been added to supplementary table 1, where IQR scores are presented. |
Supplementary table 1 |
|
7. About the hypotheses, what were they, if any, and what aspects of the literature justified them? |
Thank you. Specific hypothesis has been added Justification is provided in page 7
|
Pg 7 and 8 |
|
8. About the discussion, at the beginning, more could be written on the specific results and their interpretations, instead of writing generally about resilience. This would mirror the specific hypotheses made. |
Thank you. Discussion starts with hypothesis discussion |
Pg 21 |
Thanking you
Yours sincerely
Authors
Reviewer 3 Report
It is a very interesting study and it is a well written manuscript. I would just like to congratulate authors for their work.
Author Response
To
The reviewers
Children
Re: Manuscript ID: children-1973057
Title: Early childhood stimulating environment predicts later childhood resilience in an Indian longitudinal birth cohort study
Dear reviewer,
Thank you for reviewing our manuscript titled “Early childhood stimulating environment predicts later childhood resilience in an Indian longitudinal birth cohort study” and your encouraging words.
Thanking you
Yours sincerely
Authors
Round 2
Reviewer 1 Report
no other comments